# Integrating Multi-Omics Analysis for Enhanced Diagnosis and Treatment of Glioblastoma: A Comprehensive Data-Driven Approach

**DOI:** 10.3390/cancers15123158

**Published:** 2023-06-12

**Authors:** Amir Barzegar Behrooz, Hamid Latifi-Navid, Simone C. da Silva Rosa, Maciej Swiat, Emilia Wiechec, Carla Vitorino, Rui Vitorino, Zahra Jamalpoor, Saeid Ghavami

**Affiliations:** 1Trauma Research Center, Aja University of Medical Sciences, Tehran 14117-18541, Iran; am.barzegar.behrooz@gmail.com; 2Department of Molecular Medicine, National Institute of Genetic Engineering and Biotechnology, Tehran 14977-16316, Iran; hlatifin@gmail.com; 3Department of Human Anatomy and Cell Science, University of Manitoba College of Medicine, Winnipeg, MB R3E 3P5, Canada; simone.dasilvarosa@umanitoba.ca; 4Faculty of Medicine in Zabrze, University of Technology in Katowice, 41-800 Zabrze, Poland; maciejswiat@gmail.com; 5Division of Cell Biology, Department of Biomedical and Clinical Sciences, Linköping University, 58185 Linköping, Sweden; emilia.wiechec@liu.se; 6Coimbra Chemistry Coimbra, Institute of Molecular Sciences-IMS, Department of Chemistry, University of Coimbra, 3000-456 Coimbra, Portugal; 7Faculty of Pharmacy, University of Coimbra, 3000-456 Coimbra, Portugal; 8Department of Medical Sciences, Institute of Biomedicine iBiMED, University of Aveiro, 3810-193 Aveiro, Portugal; rvitorino@ua.pt; 9UnIC, Department of Surgery and Physiology, Faculty of Medicine, University of Porto, 4200-319 Porto, Portugal; 10Biology of Breathing Theme, Children Hospital Research Institute of Manitoba, University of Manitoba, Winnipeg, MB R3T 2N2, Canada; 11Research Institute of Oncology and Hematology, Cancer Care Manitoba-University of Manitoba, Winnipeg, MB R3T 2N2, Canada

**Keywords:** glioblastoma, biomarker selection, metabolomics, pathway analysis, personalized therapy, network analysis, inflammationomics, autophagy

## Abstract

**Simple Summary:**

The most prevalent and lethal primary brain tumor, glioblastoma multiforme (GBM), exhibits fast growth and widespread invasion and has a poor prognosis. The recurrence and mortality rates of GBM patients are still significant due to the intricacy of their molecular process. Therefore, screening GBM biomarkers is urgently required to demonstrate the therapy impact and enhance the prognosis. The findings of this study revealed 11 genes (*UBC*, *HDAC1*, *CTNNB1*, *TRIM28*, *CSNK2A1*, *RBBP4*, *TP53*, *APP*, *DAB1*, *PINK1*, and *RELN*), five miRNAs (has-mir-221-3p, hsa-mir-30a-5p, hsa-mir-15a-5p, has-mir-130a-3p, and hsa-let-7b-5p), six metabolites (HDL, N6-acetyl-L-lysine, cholesterol, formate, N, N-dimethylglycine/xylose, and X2. piperidinone), and 15 distinct signaling pathways that are essential for the development of GBM disease. The top genes, miRNAs, and metabolite signatures identified in this study may be used to develop early diagnosis procedures and construct individualized therapeutic approaches to GBM.

**Abstract:**

The most aggressive primary malignant brain tumor in adults is glioblastoma (GBM), which has poor overall survival (OS). There is a high relapse rate among patients with GBM despite maximally safe surgery, radiation therapy, temozolomide (TMZ), and aggressive treatment. Hence, there is an urgent and unmet clinical need for new approaches to managing GBM. The current study identified modules (MYC, EGFR, PIK3CA, SUZ12, and SPRK2) involved in GBM disease through the NeDRex plugin. Furthermore, hub genes were identified in a comprehensive interaction network containing 7560 proteins related to GBM disease and 3860 proteins associated with signaling pathways involved in GBM. By integrating the results of the analyses mentioned above and again performing centrality analysis, eleven key genes involved in GBM disease were identified. ProteomicsDB and Gliovis databases were used for determining the gene expression in normal and tumor brain tissue. The NetworkAnalyst and the mGWAS-Explorer tools identified miRNAs, SNPs, and metabolites associated with these 11 genes. Moreover, a literature review of recent studies revealed other lists of metabolites related to GBM disease. The enrichment analysis of identified genes, miRNAs, and metabolites associated with GBM disease was performed using ExpressAnalyst, miEAA, and MetaboAnalyst tools. Further investigation of metabolite roles in GBM was performed using pathway, joint pathway, and network analyses. The results of this study allowed us to identify 11 genes (*UBC*, *HDAC1*, *CTNNB1*, *TRIM28*, *CSNK2A1*, *RBBP4*, *TP53*, *APP*, *DAB1*, *PINK1*, and *RELN*), five miRNAs (hsa-mir-221-3p, hsa-mir-30a-5p, hsa-mir-15a-5p, hsa-mir-130a-3p, and hsa-let-7b-5p), six metabolites (HDL, N6-acetyl-L-lysine, cholesterol, formate, N, N-dimethylglycine/xylose, and X2. piperidinone) and 15 distinct signaling pathways that play an indispensable role in GBM disease development. The identified top genes, miRNAs, and metabolite signatures can be targeted to establish early diagnostic methods and plan personalized GBM treatment strategies.

## 1. Introduction

Glioblastoma (GBM) is the most common high-grade primary malignant brain tumor with a poor prognosis [1,2]. It is urgently necessary to develop new therapeutic strategies for GBM with approved treatments due to its poor survival rates [3,4]. Following decades of basic science investment in GBM, innovative clinical trials currently utilize improved genetic and epigenetic profiling to treat the disease [5,6]. Proneural, neural, classical, and mesenchymal are the four molecular subtypes of GBM [7]. The complex genetic profile of GBM is revealed by multi-omics studies of the Cancer Genome Atlas Research Network (TCGA), the Chinese Glioma Genome Atlas (CGGA), and other databases. Genetic alterations, gene transcription, and DNA methylation are molecular markers to determine prognosis and therapy selection. Molecular subtype signatures with higher resolution are essential for more effective personalized therapy [8].

A treatment for GBM can be developed if a gene signature can be identified [9]. This can be useful for diagnosis, treatment, prognosis prediction, and drug development. By analyzing the differential gene expression of astrocytomas or non-GBM gliomas, the researchers could identify a 33-gene signature of GBM. The 33 discovered signature genes included the downregulated genes *CHST9*, *CSDC2*, *ENHO*, *FERMT1*, *IGFN1*, *LINC00836*, *MGAT4C*, *SHANK2*, and *VIPR2*, as well as the overexpressed genes *COL6A2*, *ABCC3*, *COL8A1*, *FAM20A*, *ADM*, *CTHRC1*, *PDPN*, *IBSP*, *MIR210HG*, *GPX8*, *MYL9*, and *PDLIM4*. CELLO2GO’s protein functional analysis indicates that these signature genes are implicated in various biological processes. These processes include cell proliferation, adhesion, signal transduction, and the formation of anatomical structures. Many of these genes were annotated as being sensitive to stress [10]. Historically, GBMs were considered among the most heterogeneous tumors due to their diverse cellular organization and histological appearance. In addition to Telomerase Reverse Transcriptase (TERT) promoter mutations, they commonly carry copy number changes in chromosomes 7 and 10 (+7/−10). Genetic changes such as amplifying the Epidermal Growth Factor Receptor (EGFR), Platelet-Derived Growth Factor Receptor Alpha (PDGFRA), and Cyclin-Dependent Kinases 4 and 6 (CDK4/6), deletions or inactivating mutations of TP53, Phosphatase and Tensin Homolog (PTEN), Neurofibromin 1 (NF1), and CDKN2A/B could induce the variability of tumors among GBM patients [11].

Besides the crucial role that gene signatures play in GBM pathogenesis, their miRNAs can also play a pivotal role in the disease. As a result of a comprehensive, integrated analysis of microarray data, it was possible to differentiate GBM from other CNS malignancies. A total of 176 samples from 118 individuals diagnosed with GBM were included in the study for identifying dysregulated miRNAs. The only associations with GBM were found for the miRNAs hsa-miR-21-3p, hsa-miR-338-5p, hsa-miR-485-5p, hsa-miR-491-5p, and hsa-miR-1290. This characteristic was thoroughly described, focusing on tumor invasion, progression, and patient survival. Therefore, these five naturally occurring molecules exhibit differential expression in GBM and are proposed as prospective therapeutic targets. They affect various genes implicated in important signaling pathways, such as MAPK/ERK, calcium, PI3K/AKT, mTOR, and Wnt [12].

Additionally, the integrating bioinformatics and clinical analyses demonstrated the potential use of miR-1224-5p, as a prognostic and therapeutic biomarker in GBM [13]. Based on the results of another study, *PLK1*, *CCNA2*, *CCNB2*, and *AURKA* were selected as potential diagnostic marker genes based on crosstalk genes in the KEGG, PPI network, and WGCNA studies. According to the survival study, a low overall survival (OS) rate was substantially correlated with increased mRNA expression of *PLK1*, *CCNA2*, and *AURKA*. In particular, it was discovered that hsa-let-7b-5p functions as a key miRNA, controlling potential glioma-related genes. It was verified that hsa-let-7b-5p could obstruct glioma cell motility, invasion, and cell cycle [14].

This study contributes to a deeper understanding of the mechanistic basis of GBM disease. Based on phenotypic insight, transcriptomics, metabolomics, and proteogenomics are considered multi-omics approaches that contribute to discovering biomarkers for GBM diagnosis/therapeutics/prognostics. Our findings provide a new point of reference for the prognostic prediction of GBM and contribute to an in-depth understanding of the molecular mechanisms in GBM development. Furthermore, these novel signature genes might be exploited as therapeutic targets for GBM.

## 2. Materials and Methods

### 2.1. Data Integration Approaches

In our study, we did not use statistical methods for integrative analysis. This approach is appropriate when integrating different types of data together and then analyzing them. As part of the integration process, a sequential approach was first taken to analyze the protein network of different strategies. A regulatory network was constructed from the obtained proteins to identify important miRNAs. Following that, metabolic networks were analyzed separately and in conjunction with the important proteins identified.

Additionally, the type of enrichment method used is specified in the web server method. Since the approach used in this study was not one of machine learning, integrative statistical methods, variable selection methods (supervised, unsupervised, and semi-supervised), and Bayesian variable selection were not used. Moreover, stability analysis of the identified omics features in the current study was not included. Stability and validity are discussed to assess the reliability of the selected set of features and variables. One possible measure of stability which can be used in research is the measurement of the consistency of the obtained features across different data sets and platforms. To select features, one must consider whether the model is supervised, unsupervised, or semi-supervised. Since this study did not use supervised, unsupervised, or semi-supervised methods, the selection of features did not occur, so the stability was checked [15].

### 2.2. Disease Module Identification

This study found GBM-related disease modules using the NeDRex plugin version 1.0.0 (https://nedrex.net/ (accessed on 13 January 2022)) implemented in the Cytoscape platform (version 3.7.2). This investigation used two different NeDRex algorithms: MuST (Multi-Steiner trees) and DIAMOnD (DIseAse MOdule Detection). To extract a connected subnetwork engaged in the disease pathways, MuST combines a variety of approximate Steiner tree calculations that are not all unique [16,17]. Based on the idea that the connectivity importance for known disease proteins is highly distinctive, the DIAMOnD algorithm determines the disease module surrounding a set of known disease genes or proteins (seeds) [18]. The genes collected from various algorithms were each given special consideration. A summary of the data sources used in this study is presented in Table 1. Under the heading “Set of genes obtained from the two algorithms of MuST and DIAMOnD are listed and used in the study” (122 and 305 genes, respectively) (Appendix A), the genes collected from various algorithms were each given special consideration. A schematic figure of the bioinformatics approaches designed in this study is given in Figure 1.

### 2.3. GBM-Related Protein–Protein Interaction Network

Several databases, including DisGeNET (https://www.disgenet.org (accessed on 23 April 2021)) [19,20], STRING (https://string-db.org (accessed on 12 August 2021)) [21], KEGG (https://www.genome.jp/kegg) [22], and the Human microRNA Disease Database (HMDD) (https://www.cuilab.cn/hmdd (accessed on 27 March 2019)) [23], were used to identify significant proteins associated with GBM disease. Four keywords and disease IDs were used to extract data from DisGeNET, including GBM multiforme (C1621958), brain GBM (C0349543), brain stem GBM (C1332610), and GBM multiforme, somatic (C4016231). The STRING database was checked using GBM multiforme and brain GBM multiforme keywords. The KEGG database was used to extract information about the pathways involved in the development of GBM disease, including glioma (hsa05214 (accessed on 26 September 2006)), the mTOR signaling pathway (hsa04150 (accessed on 16 March 2006)), the p53 signaling pathway (hsa04115 (accessed on 24 July 2007)), the cell cycle (hsa04110 (accessed on 3 March 2023)), cytokine–cytokine receptor interaction (hsa04060 (accessed on 21 October 2020)), and the signaling pathways of calcium (hsa04020 (accessed on 18 May 2023)), ErbB (hsa04012 (accessed on 4 April 2007)), and MAPK (hsa04010 (accessed on 11 April 2023)). The acquired results were combined for network reconstruction in order to provide a list of 7560 proteins (Appendix A). In the next step, the protein–protein interaction network between these 7560 cases was reconstructed using the GeneMANIA plugin [24] and Cytoscape software (version 3.7.2). The following step involved a centrality analysis with various characteristics, including degree, closeness, betweenness, centroid, eigenvector, bridge, and eccentricity [25]. The first 20 proteins from each centrality with the highest scores were chosen (Appendix A) and combined to determine the most important proteins. Consequently, 48 essential proteins were found and selected for additional examination after processing 7560 initial entries (Appendix A).

### 2.4. Network Reconstruction of GBM-Related Signaling Pathways

A third analysis level was also established to learn about GBM disease and determine the essential proteins. Two techniques were established: (i) a literature review on genes associated with epithelial–mesenchymal transition (EMT), cytoskeleton remodeling, autophagy, secretory autophagy, and metabolism; and (ii) compiling the genes from 78 distinct signaling pathways. Table 2 lists the names of these pathways. The data from steps i and ii were combined in the following step to create a different list of 3860 genes (Appendix A). In the next step, centrality analysis was performed similarly to the previous step (Appendix A), and 20 proteins with the highest scores from each centrality were selected and combined. After processing 3860 initial entries, 35 essential proteins were discovered and selected for further analysis (Appendix A).

### 2.5. Combining the Findings from the Aforementioned Four Stages of Research and Integrated Database

The final gene list and gene network for GBM disease were developed by combining four earlier study methodologies: MuST algorithm (122 genes), DIAMOnD algorithm (305 genes), glioblastoma-related protein–protein interaction network (48 genes), and the network analysis of signaling pathways associated with GBM (35 genes). In the end, 351 genes (Appendix A) were integrated, and a network was reconstructed using the GeneMANIA plugin and Cytoscape software (version 3.7.2). Similar to the previous steps, we performed a centrality analysis and selected five proteins with the highest scores from each centrality (Appendix A). Using the results of this stage, 11 essential genes (Table 3) related to GBM disease were identified, and a miRNA-gene regulatory network was drawn via NetworkAnalyst (https://www.networkanalyst.ca/ (accessed on 10 August 2020)) [26,27] and miRTarBase v8.0 [28]. An analysis of miRNA centrality was conducted using NetworkAnalyst based on degree and betweenness. Our findings indicated that five miRNAs are essential based on the characteristics of the network based on integration results (Table 4).

### 2.6. Study of Eleven Critical Proteins in Normal Brain and Brain Tumor Expression Datasets

To ensure the expression of eleven critical proteins in the normal brain tissue and also investigate the mRNA expression (Log2) of these genes in eleven different situations (non-tumor, GBM, wild-type, mutant, primary, secondary, recurrent, classical, mesenchymal, neural, and proneural), we used ProteomicsDB (https://www.proteomicsdb.org/ (accessed on 13 September 2019)) [29,30] and Gliovis (http://gliovis.bioinfo.cnio.es (accessed on 31 October 2016)) [31] as unique web-based tools to expeditiously access data related to brain research, respectively. The Gliovis database can also be used to investigate gene expression correlations. In ProteomicsDB, transcriptomic data from the Human Protein Atlas (https://www.proteinatlas.org/ (accessed on 11 April 2016)) [32] and BGEE (https://bgee.org/ (accessed on 6 July 2016)) [33] can be integrated. At the same time, raw expression data in the Gliovis database came from various sources: ArrayExpress (https://www.ebi.ac.uk/arrayexpress/, accessed on 31 October 2016) [34], Gene Expression Omnibus (http://www.ncbi.nlm.nih.gov/geo/, accessed on 31 October 2016) [35], and Firebrowse (http://firebrowse.org, accessed on 31 October 2016) [36]. Moreover, by analyzing the TCGA database implemented in Gliovis, we evaluated the expression patterns of the identified genes, the correlation between them, and their impact on disease development mechanisms [31]. Furthermore, OSgbm (http://bioinfo.henu.edu.cn/GBM/GBMList.jsp (accessed on 31 April 2018)) was used to measure survival analysis [37].

### 2.7. Identification of Significant Metabolites and SNPs That Interact with Eleven Essential Genes

We identified significant SNPs and metabolites, interacting with eleven essential genes using the mGWAS-Explorer database (https://www.mgwas.ca/ (accessed on 15 July 2022)) [38], a user-friendly web-based tool that connects SNPs, metabolites, genes, and their known disease relationships using sophisticated network visual analytics. To identify the role of metabolites in glioma and GBM in greater depth, we extracted the list of essential metabolites (182 cases) (Appendix A) from recent studies [39,40]. In the next step, pathway, joint-pathway, and network analyses were performed using the MetaboAnalyst 5.0 (https://www.metaboanalyst.ca/ (accessed on 30 October 2019)) database [41,42].

### 2.8. Enrichment Analysis

Gene ontology and pathway enrichment analyses were carried out using the ExpressAnalyst (https://www.expressanalyst.ca/ (accessed on 7 May 2022)) [26], microRNA enrichment analysis and annotation (miEAA) (https://www.ccb.uni-saarland.de/mieaa_tool/ (accessed on 31 December 2019)) [43,44], and MetaboAnalyst 5.0 (https://www.metaboanalyst.ca/ (accessed on 30 October 2019)) [41] databases. To better understand the results, FDR < 0.05 was used to interpret the analysis outcomes.

## 3. Results

### 3.1. The Network Obtained from the NeDRex Plugin to Identify Disease Modules

The proteins in GBM were identified by delineating the disease network using the NeDRex plugin (Figure 2). In the following, step disease modules were determined by two distinct algorithms (MuST and DIAMOnD). The MuST algorithm identified five disease modules around the essential genes *MYC*, *EGFR*, *PIK3CA*, *SUZ12*, and *SPRK2*. IRAK1, PTK2, and BMI1 also represent bridging roles between MYC-EGFR, EGFR-PIK3R1, and SUZ12-SPRK2, respectively (Figure 3). The DIAMOnD algorithm identified only one module with a high number of genes (305 genes) compared to the MuST algorithm (122 genes).

### 3.2. miRNA-Gene Regulatory Network Analysis

The interaction network of the eleven essential proteins identified with the corresponding miRNAs is shown in Figure 4. After the centrality analysis, it was inevitable to investigate the interaction of the five prominent miRNAs (hsa-mir-221-3p, hsa-mir-30a-5p, hsa-mir-15a-5p, hsa-mir-130a-3p, and hsa-let-7b-5p) with eleven identified genes (Figure 5). The two miRNAs (hsa-mir-221-3p and hsa-mir-30a-5p) showed a higher degree and betweenness of the centrality levels, and their targets (TP53, CTNNB1, UBC, TRIM28, and HDAC1) seem to play a more critical role in GBM than others.

### 3.3. Analyzing the Status of Identified Gene Expression in Healthy and Malignant Brain Tissue

Using microarray and RNA-Seq data, transcriptomics results from proteomicsDB revealed that all eleven identified genes were expressed in healthy brain tissue (Figure 6). In addition, Gliovis’ findings showed that, except for *CSNK2A1*, the remaining ten genes displayed a significant variation between the GBM and the non-tumor condition. Six genes represent an increased expression in the GBM state, whereas four genes were downregulated. Furthermore, all eleven specific genes were altered during the primary stage of the tumor (Table 5). As a consequence of the correlation analysis, ten positive correlations were found between UBC (APP), HDAC1 (TP53, RBBP4, TRIM28, and CTNNB1), RBBP4 (CTNNB1 and TP53), and TRIM28 (TP53, RBBP4, and CSNK2A1), and eight negative correlations were found between RBBP4 (APP), PINK1 (TRIM28, RBBP4, TP53, and HDAC1), and DAB1 (UBC, HDAC1, and TP53) (Figure 7). OSgbm survival analysis indicated that HDAC1 (HR = 1.327, *p* < 0.029) and RELN (HR = 0.752, *p* < 0.029) significantly affected overall survival. The study’s results demonstrated that HDAC1 and RELN could be considered diagnostic biomarkers and have prognostic significance for GBM (Figure 8).

### 3.4. Enrichment Analysis

Eleven significant GSEA results were categorized into four axes. Negative control of the cellular process (FDR = 0.00165), negative control of the cell cycle (FDR = 0.00165), protein phosphorylation (FDR = 0.00165), negative control of the biological process (FDR = 0.00165), and negative control of the apoptotic process (FDR = 0.00165) were significant in the biological process (Appendix A). Enzyme binding (FDR = 0.000316), positive transcription regulation (FDR = 0.0055), DNA-dependent negative transcription regulation (FDR = 0.0188), and DNA-dependent transcription from RNA polymerase II promoter (FDR = 0.0315) all played a crucial role in molecular function (Appendix A). Nuclear chromatin (FDR = 4.04 × 10^−6^), nuclear chromosome part (FDR = 3.01 × 10^−5^), nuclear chromosome (FDR = 3.63 × 10^−5^), chromatin (FDR = 3.63 × 10^−5^), and histone deacetylase complex (FDR = 0.000161) in cellular components (Appendix A) were indispensable, as well as mitophagy (FDR = 0.00978) and the Wnt signaling pathway (FDR = 0.046), in the KEGG pathway enrichment analysis (Appendix A). Additionally, miEAA-related results showed that fatty acid biosynthesis (0.0078415), galactose metabolism (0.0078415), mucin-type O-glycan biosynthesis (0.0233921), and autophagy (0.0264018) might also be crucial in GBM disease (Appendix A). (Figure 9 and Figure 10). 

The third level of analysis was performed using the metabolites, KEGG, and SMPDB databases [45]. The three main FDR-based pathways obtained from KEGG (Appendix A) were aminoacyl-tRNA biosynthesis (1.39 × 10^−8^), arginine biosynthesis (3.94 × 10^−7^), alanine, aspartate, and glutamate metabolism (2.03 × 10^−6^). In comparison, three other main pathways were identified using the SMPDB databases (Appendix A): glutamate metabolism (9.34 × 10^−9^), urea cycle (9.34 × 10^−9^), and arginine and proline metabolism (3.24 × 10^−8^) (Figure 11).

### 3.5. Metabolic Pathway Analysis

The outcomes of the metabolic pathway enrichment analysis and pathway topology analysis were combined for this investigation. The results were divided into four categories according to the factors considered, as shown in Table 6.

The results were analyzed and summarized based on three criteria: 1. The selection of the first five cases in each of the centralities—relative betweenness centrality (R-b C) and out-degree centrality (O-d C)—based on the highest score obtained in the impact parameter. 2. In the next step, the results were classified based on FDR, and five cases (phenylalanine, tyrosine, and tryptophan biosynthesis; synthesis and degradation of ketone bodies; one-carbon (1C) pool by folate; trehalose degradation; and glycerol phosphate shuttle) were excluded because they were not significant in the metabolic pathway analysis (red color). 3. The common items in both cases (R-b C and O-d C) and databases (KEGG and SMPDB) were considered vital and further discussed. Two items were obtained from the KEGG database (nitrogen metabolism, alanine, aspartate, and glutamate metabolism) and three items were obtained from the SMPDB database (alanine metabolism, aspartate metabolism, and malate-aspartate shuttle) (Table 7 and Appendix A).

### 3.6. Joint Pathway Analysis

We used joint pathway analysis to simultaneously analyze eleven identified essential genes and 182 distinctive metabolites within metabolic pathways. Three types of results were obtained based on the topology measure used (Table 8). The results were analyzed and summarized based on four criteria: 1. The selection of the first 10 cases in each of the centralities based on the highest score obtained in the impact parameter. 2. In the next step, the results were classified based on FDR, and five cases (1C pool by folate, p53 signaling pathway, phosphatidylinositol signaling system, longevity regulating pathway, and mitophagy-animal) were excluded because they were not significant in the joint pathway analysis. 3. Considering the results obtained from all three of the centralities, the two items observed in all three (citrate cycle and arginine biosynthesis) were selected and further discussed (Table 9 and Appendix A). The results were combined in one table to obtain a comprehensive overview of the pathway enrichment analysis at different levels (eleven essential proteins, five miRNAs, and 182 metabolites), pathways, and joint pathway analyses (Table 10).

### 3.7. Gene–Metabolite Interaction Network

Drawing the interaction network between the genes and metabolites showed that the *APP* and *TP53* genes are related to each other through five factors (adenosine triphosphate, ADP, glycerol, L-glutamic acid, and L-lysine). Moreover, the connection between RELN, CTNNB1, and CSNK2A1 and APP was shown through Gamma-aminobutyric acid (GABA), palmitic acid, and glycerol, respectively (Figure 12).

### 3.8. Identification of SNPs-Related Metabolites and Genes

Using eleven critical genes as input data in the mGWAS-Explorer database, six metabolites and 23 SNPs were identified (Table 11 and Appendix A). Additionally, we determined the top 25 SNPs via 182 metabolites as input data in the MetaboAnalyst 5.0 database (Figure 13).

## 4. Discussion

Gene sequencing studies have made it possible to learn more about GBM genetics and epigenetics in recent years [46]. A practical approach for diagnosing and treating GBM is to use molecular biomarkers. The current study used genes, miRNAs, and metabolites to develop a panel of predictive biomarkers for GBM.

Based on the biological pathways, the identified genes in the GBM biomarker panel were classified into different signaling pathways such as the AKT (MYC, BMI1, EGFR, PIK3CA, PTK2, and UBC), the inflammation (IRAK1 and APP), the P53 (HDAC1, P53, and TRIM28), the WNT (CTNNB1), and the mitochondrial signaling pathways (DAB1, PINK1, and RELN). AKT regulates angiogenesis and metabolism [47]. Numerous human tumors have elevated levels of *MYC*, including GBM [48]. There is a correlation between Myc expression and glioma grade [49,50], and it was shown that inhibiting Myc in gliomas reduces proliferation and increases apoptosis [51]. The *BMI-1* is another important gene related to the AKT signaling pathway. The *BMI1* gene belongs to the polycomb group (PcG) gene family and is a transcriptional repressor of several genes that govern cell proliferation and differentiation throughout life [52,53,54,55,56]. In GBM, the simultaneous targeting of EZH2 and BMI1 was more effective than either agent alone due to the presence of both proneural and mesenchymal GSCs [57].

There is evidence that *EGFR* overexpression is associated with more aggressive GBM phenotypes in most primary GBMs and some secondary GBMs [58]. An analysis of the TCGA GBM database uncovered a subgroup with *EGFR* amplification and *TP53* mutations. Both of these functions are almost mutually exclusive, suggesting EGFR regulates the function of wild-type *p53* (wt-*p53*). EGFR signaling inhibits the function of wt-p53 in GBM by facilitating the interaction between p53 and the DNA-dependent protein kinase catalytic subunit (DNA-PKcs) [59].

It has been reported that 6–15% of glioblastomas contain activating mutations in the *PIK3CA*. There is evidence that *PIK3CA*-activating mutations are associated with an earlier recurrence of GBM in adults and a shorter survival time [60]. Additionally, immunohistochemical analysis of most anaplastic astrocytomas and glioblastomas demonstrated a strong expression of the PTK2 protein [61]. Elevated PTK2 (focal adhesion kinase 1 (Fak1)) protein levels were detected in astrocytic gliomas [62].

In the case of GBM, ubiquitin-dependent mechanisms may be exploited as a therapeutic strategy. It was indicated that the ubiquitin system is involved in core signaling pathways, including EGFR, TGF-β, p53, and stemness-related pathways in GBM [63]. In addition, it was shown that the inhibition of ubiquitin signaling could reverse metabolic reprogramming and suppresses GBM growth. The regulation of protein stability by the ubiquitin–proteasome system (UPS) represents an important control mechanism of cell growth in various human cancers, including GBM [64].

Evidence from both in vitro and in vivo studies shows that the highest expressed form of *IRAK1* in low-grade gliomas (LGG) inhibits cell apoptosis and increases malignancy [65]. GBM is positively associated with mortality in Alzheimer’s disease (AD) [66]. There is an up-regulation of the HDAC class I isoforms HDAC1 and HDAC2 in GBM cell lines compared with non-neoplastic brain tissues [67,68]. Proliferating, migrating, and invading are inhibited in GBM cells when *HDAC1* and *HDAC2* expressions are silenced. Similarly, HDAC3 is overexpressed in aggressive glioma cell lines and is associated with poor prognosis and OS of GBM patients [69]. A selective histone deacetylase inhibitor induces autophagy and cell death in GBM cells by downregulating SCNN1A [70]. A significant correlation was found between most members of the HDAC family and glioma grade, IDH1 mutation, and 1p/19q co-deletion. Among the HDAC1-related signatures for precise prognosis prediction in glioma, HDAC1 indicates prognosis and immune infiltration [71].

GBM is commonly associated with *TP53* mutations. Approximately 84% of GBM patients exhibit dysregulation of the p53-ARF-MDM2 pathway, a finding that is confirmed in 94% of GBM cell lines adopted for in vitro assays [72,73].

Diverse cellular functions are mediated by PI3K/Akt-WNT signaling interactions in GBM, including cell proliferation, EMT, metabolism, and angiogenesis [47]. A study has shown that inhibiting WNT-CTNNB1 signaling enhances the SQSTM1 expression and sensitizes GBM cells to autophagy blockers [74]. This pathway also regulates autophagy and mitophagy [74,75,76]. Mitophagy, a selective autophagy of mitochondria, is crucial for quality control since it can efficiently degrade, remove and recycle malfunctioning or damaged mitochondria [77]. It has been demonstrated that platelet-derived growth factor (PDGF) signaling induces N6-methyladenosine (m^6^A) accumulation in GSCs to regulate mitophagy. A PDGF ligand stimulates the transcription of early growth response 1 (EGR1), which promotes the proliferation and self-renewal of GSCs by inducing methyltransferase-like 3 (METTL3). By regulating the m^6^A modification of optineurin (OPTN), the PDGF-METTL3 axis inhibits mitophagy. In GBM patients, the forced expression of OPTN mimics the inhibition of PDGF, and higher OPTN levels predict a longer survival time [78].

There were three categories of miRNAs identified in the GBM biomarker panel: proliferation (hsa-mir-221-3p), invasion (hsa-mir-15a-5p and hsa-let-7b-5p), and proliferation and invasion (hsa-mir-30a-5p and hsa-mir-130a-3p).

In GBM, miR-221/222, which targets the p53 upregulated modulator of apoptosis (PUMA), was reported to induce cell survival [79]. There is evidence that the chronic miR-221/222-mediated downregulation of MGMT may result in cells being unable to repair genetic damage. The presence of miR-221/222 oncogenic potential may improve the prognosis of GBM [80]. Furthermore, decreased EGFR and increased miR-221 were associated with increased resistance to temozolomide (TMZ) and radiotherapy in GBM [81], compared to normal brain tissues (NBTs). MiR-30a-5p is overexpressed in glioma cell lines and glioma samples, with its expression level positively correlated with tumor grade [82]. According to researchers, the Wnt/β-catenin–miR-30a-5p–NCAM regulatory axis is essential in controlling glioma cell invasion and tumorigenesis. It was shown that the Wnt/β-catenin pathway activates miR-30a-5p through the direct binding of β-catenin/TCF4 to two sites in the promoter region of miR-30a-5p. In addition, miR-30a-5p can inhibit the expression of neural cell adhesion molecule (NCAM) by directly targeting two sites in the 3′-untranslated regions (3′-UTR) of NCAM mRNA [83].

The proliferation and invasion of GBM cells are mediated by several critical molecules, such as cell adhesion molecule 1 (CADM1). CADM1 expression is decreased in GBM patients and GBM cell lines, and CADM1 overexpression inhibits the proliferation of GBM cells. According to these findings, CADM1 effectively suppresses the proliferation of GBM. MiR-15a-5p was shown to promote the proliferation and invasion of T98G GBM cells by targeting CADM1 [84,85].

In the GBM biomarker panel, five categories of metabolites were identified: lipid metabolism (cholesterol), glutamate metabolism (glutamate and GABA), tricarboxylic acid (TCA) cycle (alanine), urea cycle (arginine), and the Leloir cycle (galactose). There is a link between metabolic syndrome and several types of cancer, including GBM.

An analysis of a New Zealand cohort of GBM patients showed that metabolic syndrome is associated with reduced OS. In light of this finding, there is a greater likelihood that GBM results from metabolic pathogenesis [86]. According to studies, lipid metabolism plays a critical role in the pathogenesis of GBM. Among the members of the apolipoprotein family, apolipoprotein C1 (ApoC1) is crucial for the metabolism of both very-low-density lipoprotein (VLDL) and high-density lipoprotein (HDL) cholesterols. The pre-surgery level of serum LDL cholesterol was a prognostic factor for the outcome of patients with GBM [87].

Cholesterol is another important molecule with the potential role of repurposed drugs. Due to the discovery that many cancers, including GBM, reprogrammed the cholesterol metabolism, the cholesterol metabolism has become a promising potential target for therapy [88]. Different strategies for inhibiting cholesterol metabolism have been proposed since GBM cells require external cholesterol for survival and lipid droplets for rapid growth. The activation of liver X receptors (LXRs) inhibits cholesterol uptake, promotes cholesterol efflux, disrupts cholesterol trafficking within cells, interferes with SREBP signaling, and impedes cholesterol esterification. They may potentially counteract glial tumor growth [89,90]. It has also been demonstrated that lipid accumulation and oxidation play a role in GBM. Monounsaturated fatty acids have been found to promote GBM proliferation by modulating triglyceride metabolism [91]. A knockdown of carnitine palmitoyltransferase 1A (CPT1A), a critical enzyme in fatty acid oxidation (FAO), also reduced tumor growth and increased survival, according to in vivo studies [92].

The excitatory neurotransmitter glutamate plays a significant role in brain tumor cells’ proliferation and growth. Glutaminase produces a large amount of glutamate in glioma cells, which converts glutamate from glutamine and increases intracellular Ca^2+^ through P2 × 7Rs [93]. Moreover, high levels of glutamate have been found to cause brain edema and seizures in glioma patients. Glutamate and glutamine are linked to the proline pathway. L-proline is a multifunctional amino acid essential in the primary metabolism and physiological functions. Proline is oxidized to glutamate in the mitochondria, and the FAD-containing enzyme proline oxidase (PO) catalyzes the first step in the L-proline degradation pathway. It was shown that PO might play a regulatory role in glutamatergic neurotransmission by affecting the cellular concentration of glutamate [94]. Study results indicate that serine and glycine levels are higher in the low-nutrient regions of GBM tumors than in the other areas. A study of the metabolic and functional properties of GBM cells revealed that serine availability and 1C metabolism support glioma cells’ survival following glutamine deprivation. Serine synthesis was mediated by autophagy rather than glycolysis [95].

ATP by glycolysis and the TCA cycle are associated with oxidative phosphorylation (OXPHOS) through the breakdown of pyruvate or fatty acids to meet the growing energy demand of cancer cells. Recent studies demonstrate that SMI EPIC-0412 can effectively perturb the TCA cycle. It was shown that in combination with the cytosolic phospholipase A2 (cPLA2) inhibitor AACOCF3 and the hexokinase II (HK2) inhibitor 2-DG, SMI EPIC-0412 disrupted GBM energy metabolism for targeted metabolic therapy. ATP production significantly declined in glioma cells when treated with a monotherapy (EPIC-0412 or AACOCF3), dual therapy (EPIC-0412 + AACOCF3), or triple therapy (EPIC-0412 + AACOCF3 + 2-DG) regimen [96]. Furthermore, it has been shown that patients with GBM with high levels of glycolysis-related genes, including HK2 and PKM2, and low levels of mitochondrial metabolism-related genes, such as SDHB and COX5A, related to TCA cycle and oxidative phosphorylation (OXPHOS), have poor survival rates. In contrast to LGG, the expression levels of genes involved in the mitochondrial oxidative metabolism in GBM are markedly increased; however, they are lower than those in normal brains [97].

It has been shown that dysregulated alanine could serve as a potential predictive marker for glioma [98]. Alanine, a glucogenic amino acid, enters the metabolic stream through enzymatic conversion to pyruvate to provide energy and replenish the nutrient reservoir for rapidly proliferating tumor cells [99]. Arginine is another amino acid substrate actively metabolized by tumor cells to promote tumor growth and immunosuppression [100,101]. Arginine transporters appear abundant in GBM, as evidenced by the accumulation of byproducts of arginine metabolism [102,103]. The results indicate that the arginine metabolism is functional and may be sensitive to targeted depletion. Recent research demonstrated that pegylated human recombinant ARG1 depleted arginine in glioma cells and induced cytotoxicity [104].

GBM tumor cell proliferation depends on the availability of extracellular nutrients. As a result of inadequate tumor perfusion, glucose and glutamine are in short supply. Due to this metabolic remodeling, GBMs scavenge alternative nutrients from the tumor microenvironment to sustain their growth and proliferation. Glut3 and Glut14 are sugar transporters expressed in GBM. GBM cells can scavenge galactose (Gal) from the circulation and extracellular space as a suitable substrate for Glut3/Glut14. The Leloir pathway provides GBM cells with an alternative energy source by transporting and metabolizing Gal at physiological Glc concentrations [105].

## 5. Conclusions and Future Direction

Research on GBM has made significant progress in recent years, partly due to the availability of large-scale multi-omics databases and other data sources that allow researchers to better understand the disease’s genetic and molecular characteristics. It is important to identify genes, miRNAs, and metabolites that contribute to the progression of GBM. In our current investigations, we identified eleven important genes and five miRNAs that have a critical role in the pathogenesis of GBM. In addition, we found essential metabolites that drive GBM development. These findings highlight the importance of genes, miRNAs, and metabolites in GBM progression. 

It is critical for both basic and clinical scientists to find out which genes or proteins lead to or are associated with metabolic changes. In future, transcriptome and proteomic analyses should identify key genes and proteins contributing to inflammation in GBM progression. Understanding how inflammation influences GBM progression is critical for developing new therapeutic strategies. Our team will focus on tailoring formulations and repurposing drugs to develop diagnostic and therapeutic biomarkers in the near future and open up new avenues for the personalized treatment of patients with GBM.

## Figures and Tables

**Figure 1 cancers-15-03158-f001:**
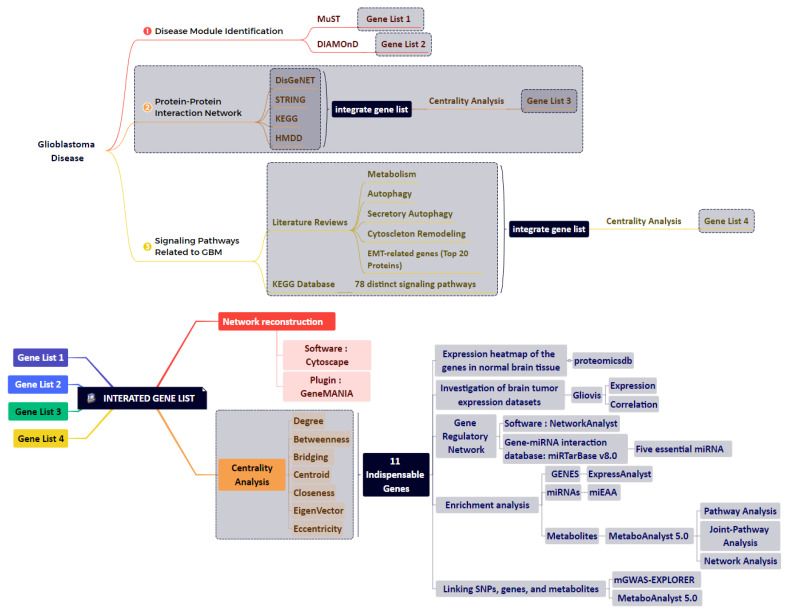
An overview of the bioinformatics approaches used in this study. We created four gene lists from several different study levels. By merging these gene lists and performing various analyses, 11 genes and five key miRNAs were identified.

**Figure 2 cancers-15-03158-f002:**
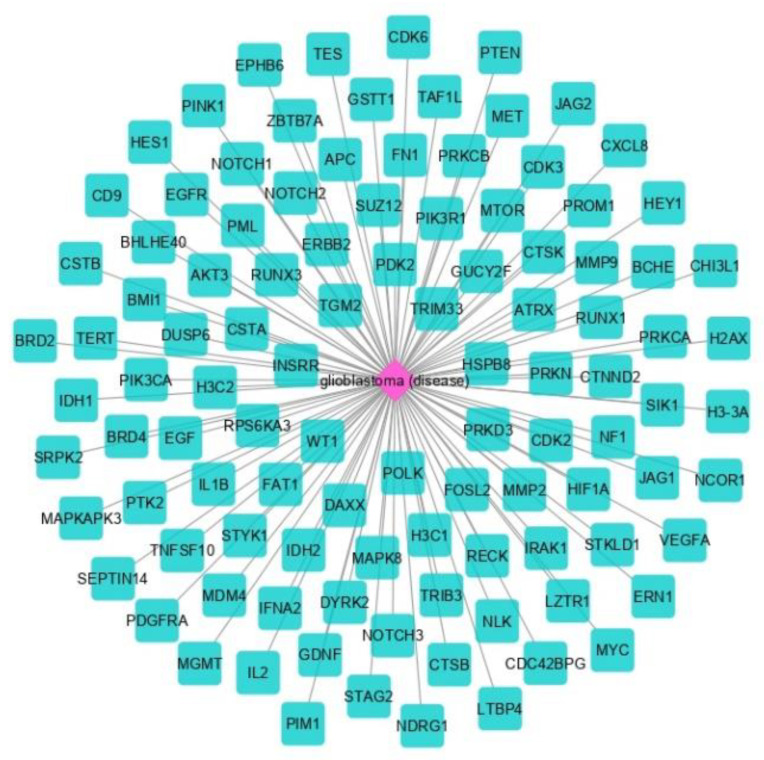
Glioblastoma-related proteins identified by the NeDRex plugin.

**Figure 3 cancers-15-03158-f003:**
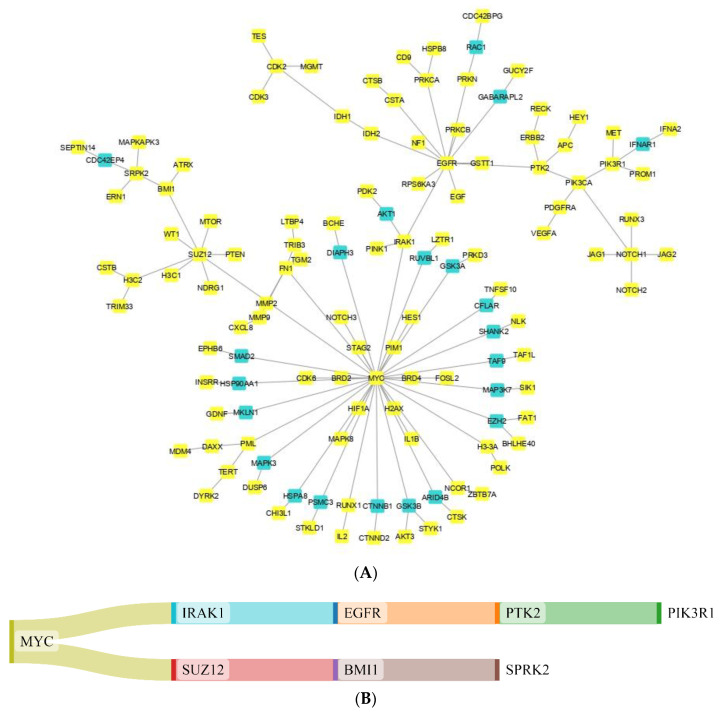
Modules of disease identified by the MuST algorithm (**A**) and genes (**B**) that interconnect them. Five essential genes were identified: *MYC*, *EGFR*, *PIK3CA*, *SUZ12*, and *SPRK2*. Additionally, *IRAK1, PTK2*, and *BMI1* represent bridging roles between the disease modules.

**Figure 4 cancers-15-03158-f004:**
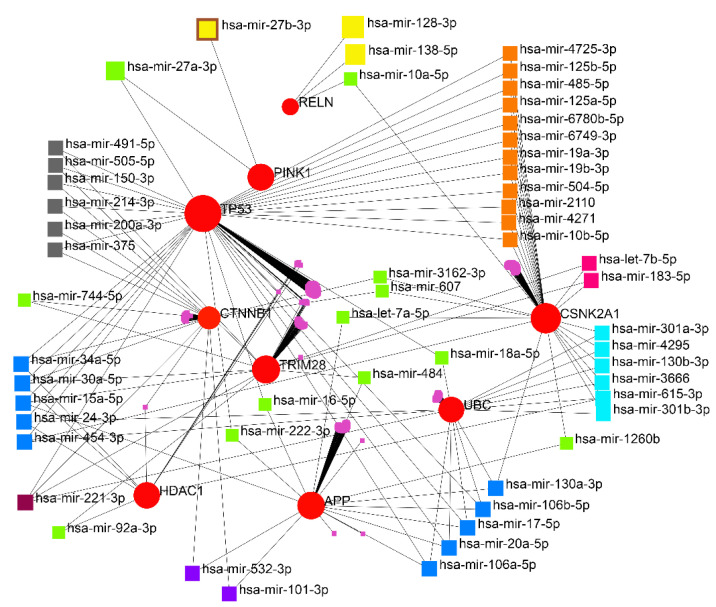
Gene regulatory network obtained from eleven identified proteins. A wide range of interacting genes-miRNAs was determined.

**Figure 5 cancers-15-03158-f005:**
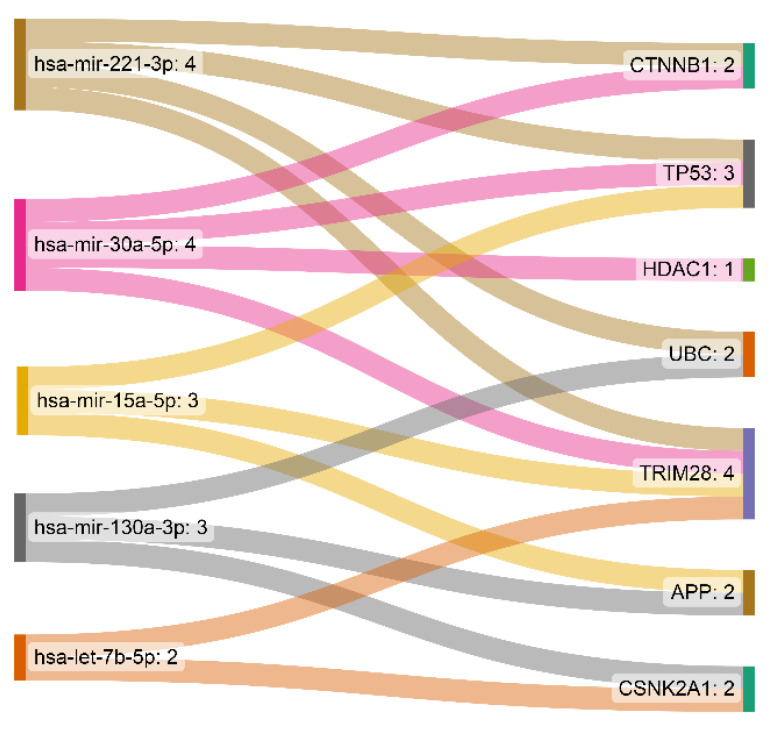
The scheme of the relationship between the five identified miRNAs and their targets. hsa-mir-221-3p and hsa-mir-30a-5p showed a higher degree and betweenness of the centrality levels.

**Figure 6 cancers-15-03158-f006:**
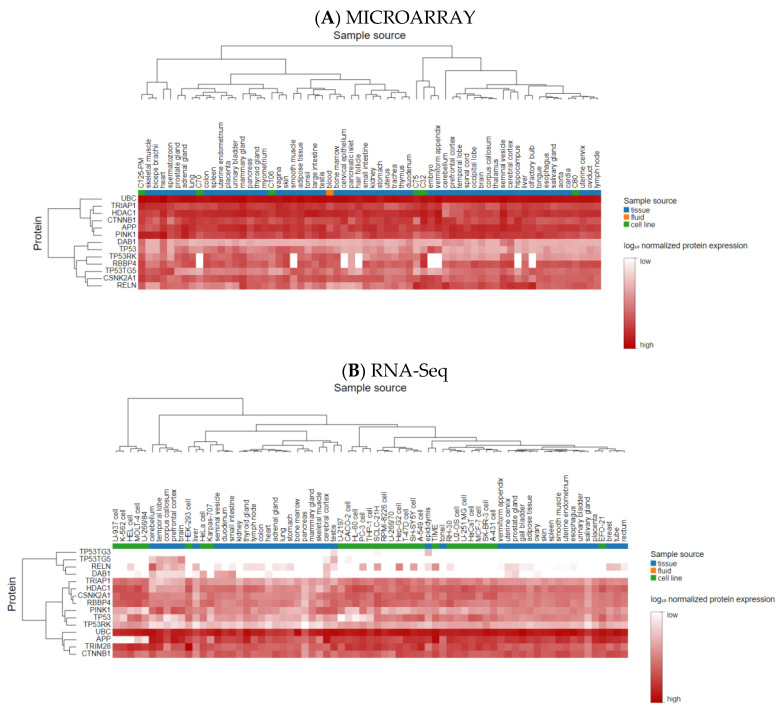
The heatmap of the expression of eleven genes in normal brain tissue (**A**) based on a microarray and (**B**) based on RNA-Seq.

**Figure 7 cancers-15-03158-f007:**
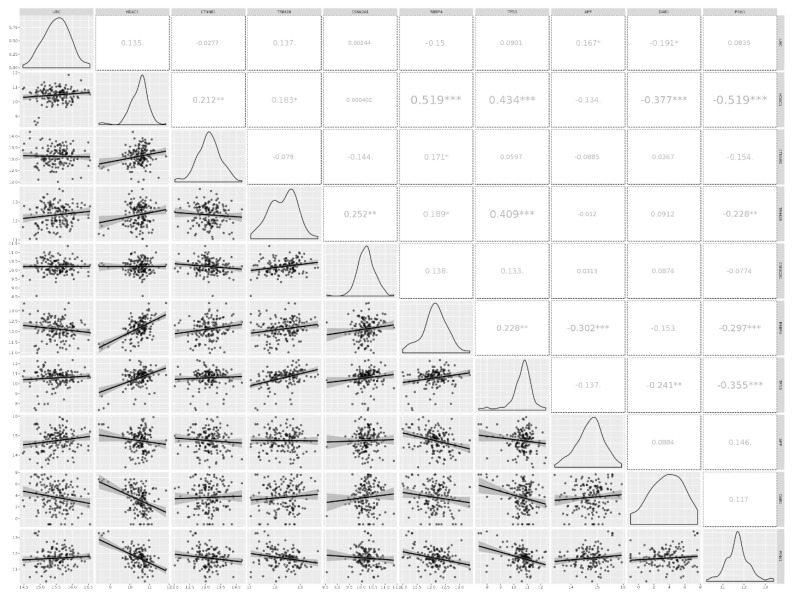
The results of correlation analysis between eleven identified genes. Ten positive correlations were found between UBC (APP), HDAC1 (TP53, RBBP4, TRIM28, and CTNNB1), RBBP4 (CTNNB1 and TP53), and TRIM28 (TP53, RBBP4, and CSNK2A1), and eight negative correlations were found between RBBP4 (APP), PINK1 (TRIM28, RBBP4, TP53, and HDAC1), and DAB1 (UBC, HDAC1, and TP53). *: *p* < 0.05, **: *p* < 0.01, ***: *p* < 0.001.

**Figure 8 cancers-15-03158-f008:**
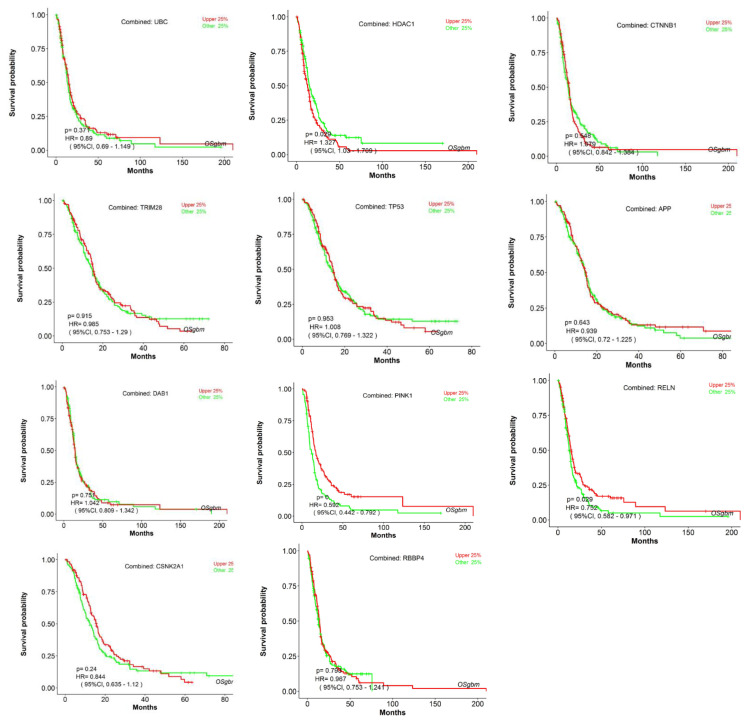
Survival analysis. Analysis of the prognostic value of identified genes using a combined cohort with pooling all datasets (Cancer Genome Atlas (TCGA), Gene Expression Omnibus (GEO), and Chinese Glioma Genome Atlas (CGGA)) together in OSgbm. The survival analysis results were presented using a Kaplan–Meier (KM) plot with a hazard ratio (HR) and log-rank *p*-value.

**Figure 9 cancers-15-03158-f009:**
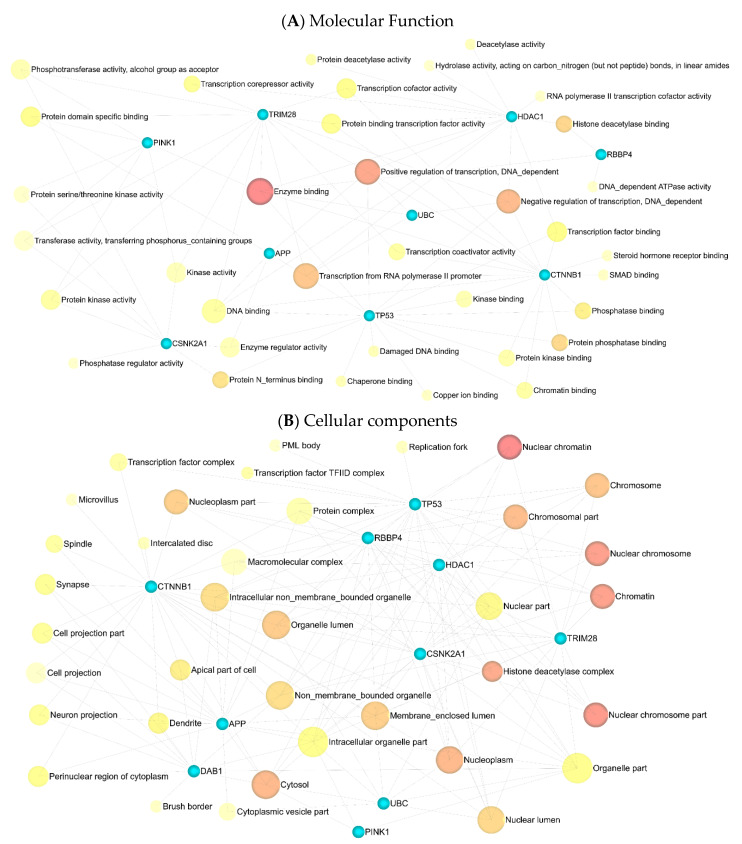
The results of gene ontology analysis between eleven identified genes: (**A**) molecular function; (**B**) cellular components. Darker shades have greater significance.

**Figure 10 cancers-15-03158-f010:**
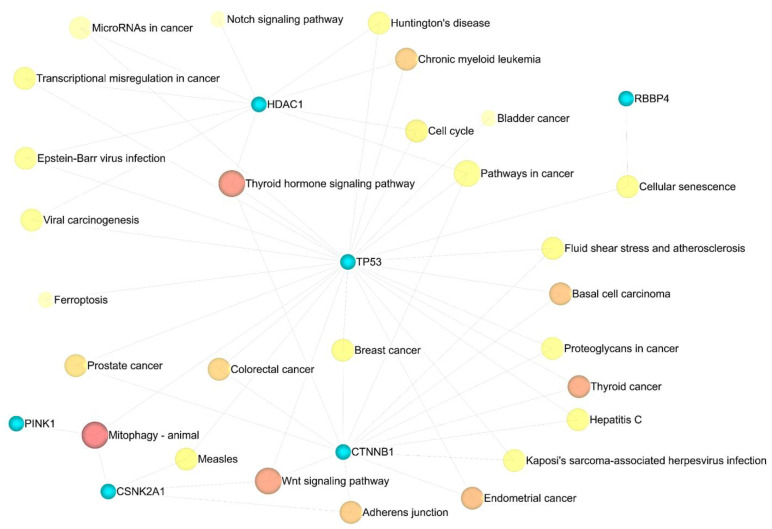
The association of the KEGG pathway enrichment analysis results with eleven identified genes. Darker shades are more significant.

**Figure 11 cancers-15-03158-f011:**
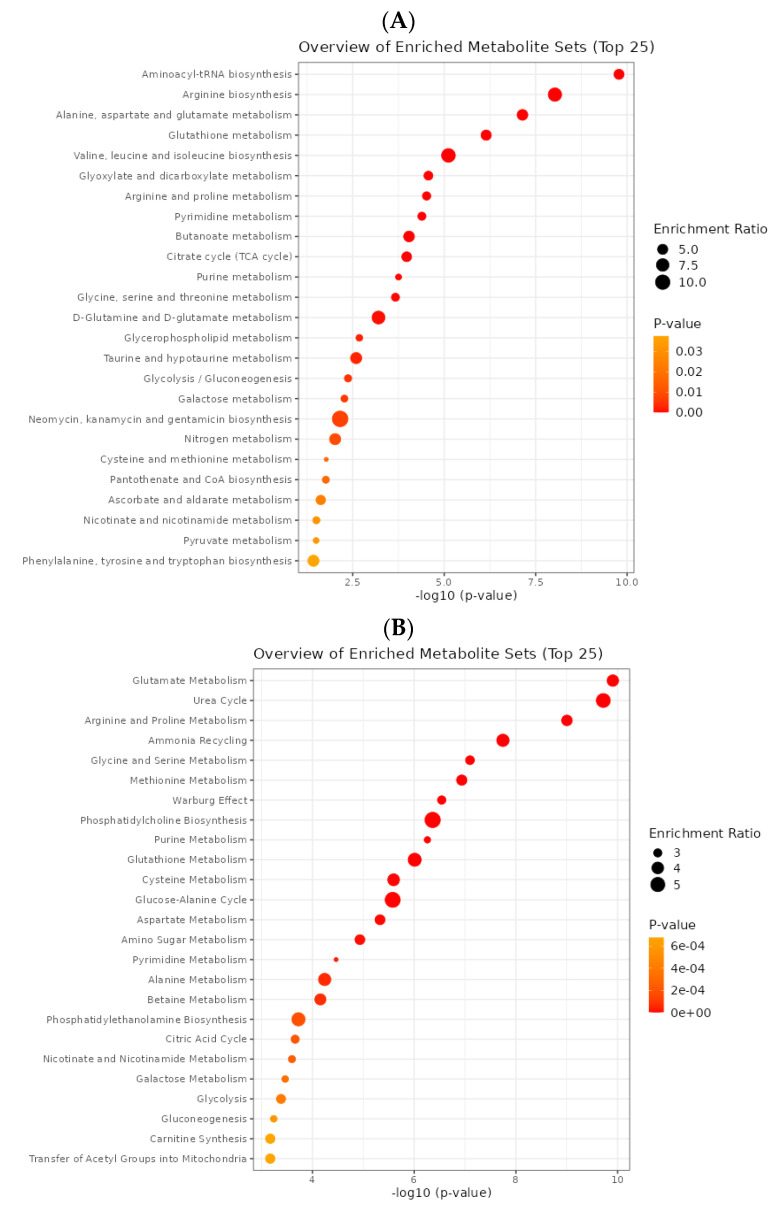
The association of metabolic pathway enrichment analysis results with 182 metabolites: (**A**) based on the KEGG database; (**B**) based on the SMPDB database.

**Figure 12 cancers-15-03158-f012:**
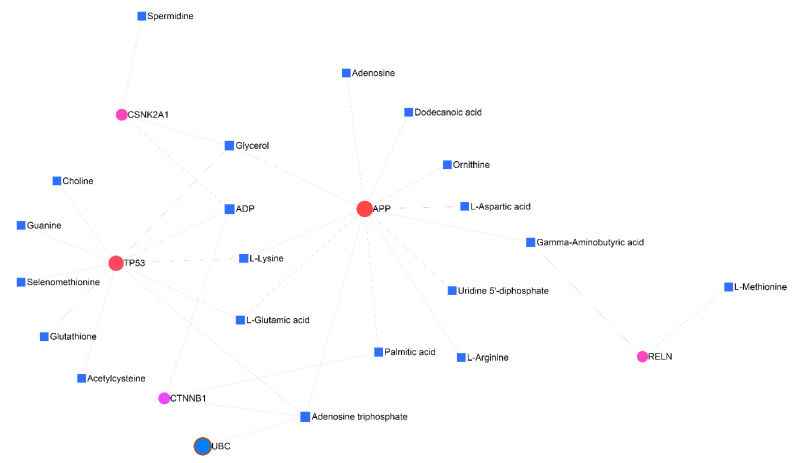
Gene–metabolite interaction network. The factors involved in the relationship between *TP53*, *CTNNB1*, *CSNK2A1*, and *RELN* with APP were further investigated.

**Figure 13 cancers-15-03158-f013:**
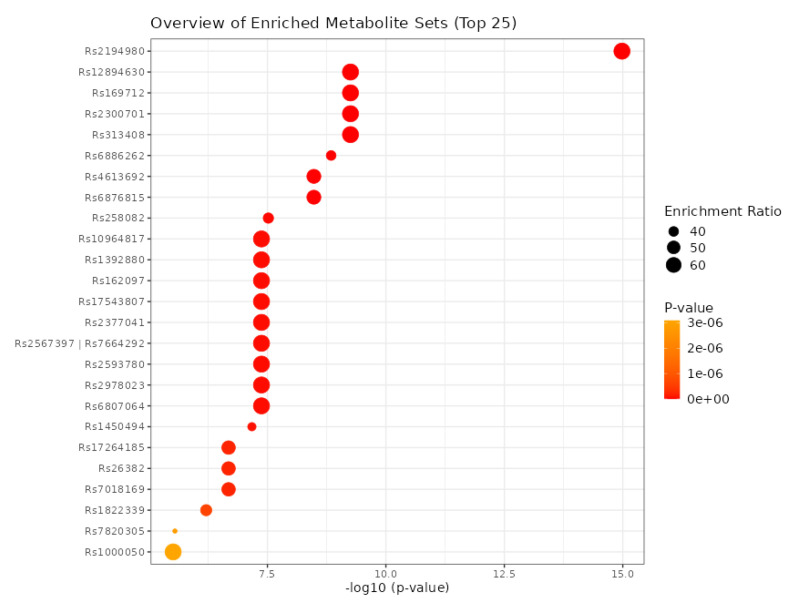
The determination of the top 25 SNPs using the MetaboAnalyst 5.0 database.

**Table 1 cancers-15-03158-t001:** Data sources for GBM multi-omics analysis.

Database Name	Type of Data	Purpose
The Cancer Genome Atlas Research Network (TCGA)	Genomic, Epigenomic, Transcriptomic	Investigate the genetic profile and molecular subtypes of GBM
NeDRex plugin version 1.0.0	Disease Module Detection	Find GBM-related disease modules in the Cytoscape platform
MuST Algorithm	Approximate Steiner Tree Calculation	Extract a connected subnetwork engaged in the disease pathways
DIAMOnD Algorithm	Disease Module Detection	Determine the disease module surrounding a set of known disease genes or proteins
STRING	Proteins	Protein–protein Interaction Networks
KEGG	Proteins	Find GBM-related proteins
HMDD	miRNAs and proteins	GBM-related miRNAs-proteins Interaction network
GlioVis	Over 6500 tumor samples of approximately 50 expression datasets of a large collection of brain tumor entities (mostly gliomas), both adult and pediatric	To analyze the correlation between identified genes based on the TCGA database
OSgbm	Transcriptome profiles and clinical information from The Cancer Genome Atlas (TCGA), Gene Expression Omnibus (GEO), and Chinese Glioma Genome Atlas (CGGA).	An online consensus survival analysis web server

**Table 2 cancers-15-03158-t002:** Data collection using 78 signaling pathways.

1	VEGF signaling pathway-hsa04370	40	TNF signaling pathway-hsa04668
2	PI3K-Akt signaling pathway-hsa04151	41	Citrate cycle (T.C.A. cycle)-hsa00020
3	Ras signaling pathway- hsa04014	42	Glycolysis/Gluconeogenesis-hsa00010
4	TGF-beta signaling pathway-hsa04350	43	Oxidative phosphorylation-hsa00190
5	HIF-1 signaling pathway-hsa04066	44	Starch and sucrose metabolism-hsa00500
6	AMPK signaling pathway-hsa04152	45	Pentose phosphate pathway-hsa00030
7	MAPK signaling pathway-hsa04010	46	Pyruvate metabolism-hsa00620
8	Rap1 signaling pathway-hsa04015	47	Insulin signaling pathway-hsa04910
9	Wnt signaling pathway-hsa04310	48	Lysosome-hsa04142
10	Notch signaling pathway-hsa04330	49	Phospholipase D signaling pathway-hsa04072
11	Hedgehog signaling pathway-hsa04340	50	Mitophagy- hsa04137
12	Hippo signaling pathway-hsa04390	51	Signaling pathways regulating pluripotency of stem cells- hsa04550
13	JAK-STAT signaling pathway-hsa04630	52	Cell adhesion molecules-hsa04514
14	Apelin signaling pathway-hsa04371	53	Cell cycle -hsa04110
15	NF-kappa B signaling pathway-hsa04064	54	ECM-receptor interaction-hsa04512
16	TNF signaling pathway-hsa04668	55	PD-L1 expression and PD-1 checkpoint pathway in cancer- hsa05235
17	FoxO signaling pathway-hsa04068	56	Pathways in cancer-hsa05200
18	Phosphatidylinositol signaling system-hsa04070	57	Transcriptional misregulation in cancer-hsa05202
19	mTOR signaling pathway-hsa04150	58	Central carbon metabolism in cancer-hsa05230
20	p53 signaling pathway-hsa04115	59	IL-17 signaling pathway-hsa04657
21	Apoptosis-hsa04210	60	Necroptosis-hsa04217
22	Ubiquitin-mediated proteolysis-hsa04120	61	Cellular senescence-hsa04218
23	Cell cycle-hsa04110	62	Chemokine signaling pathway-hsa04062
24	Regulation of actin cytoskeleton-hsa04810	63	Transcriptional misregulation in cancer-hsa05202
25	Calcium signaling pathway-hsa04020	64	ECM-receptor interaction-hsa04512
26	T cell receptor signaling pathway-hsa04660	65	Proteoglycans in cancer-hsa05205
27	Focal adhesion-hsa04510	66	Choline metabolism in cancer-hsa05231
28	Adherens junction-hsa04520	67	PD-L1 expression and PD-1 checkpoint pathway in cancer-hsa05235
29	Gap junction-hsa04540	68	Ferroptosis-hsa04216
30	Tight junction-hsa04530	69	Cholesterol metabolism-map04979
31	Arachidonic acid metabolism-hsa00590	70	Lipid and atherosclerosis-map05417
32	Autophagy-hsa04140	71	Fat digestion and absorption-map04975
33	Regulation of lipolysis in adipocytes-hsa04923	72	Vitamin digestion and absorption-map04977
34	Cytokine-cytokine receptor interaction-hsa04060	73	Aldosterone synthesis and secretion-map04925
35	Proteasome- hsa03050	74	Primary bile acid biosynthesis-map00120
36	B cell receptor signaling pathway-hsa04662	75	Cortisol synthesis and secretion-map04927
37	Complement and coagulation cascades-hsa04610	76	Bile secretion-map04976
38	Toll-like receptor signaling pathway-hsa04620	77	Ovarian steroidogenesis-map04913
39	RIG-I-like receptor signaling pathway-hsa04622	78	Steroid biosynthesis-map00100

**Table 3 cancers-15-03158-t003:** The identification of eleven critical genes through integrating results and network analysis. The list of important genes is based on their centrality. Deg: Degree, Bet: Betweenness, Bri: Bridge, Cent: Centroid, Close: Closeness, and EiVe: EigenVector. “+”: presence; “--”: absence.

Gene Name	Description	Deg	Bet	Bridg	Cent	Close	EiVe
UBC	Ubiquitin C [Source: HGNC Symbol; Acc: HGNC:12468	+	+	--	--	+	+
HDAC1	Histone deacetylase 1 [Source: HGNC Symbol; Acc: HGNC:4852	+	--	--	--	+	+
CTNNB1	Catenin beta 1 [Source: HGNC Symbol; Acc: HGNC:2514	+	--	--	--	+	+
TRIM28	Tripartite motif-containing 28 [Source: HGNC Symbol; Acc: HGNC:16384	--	+	--	--	+	+
CSNK2A1	casein kinase two alpha 1 [Source: HGNC Symbol; Acc: HGNC:2457	--	--	--	--	+	+
RBBP4	RB binding protein 4, chromatin remodeling factor [Source: HGNC Symbol; Acc: HGNC:9887	+	--	--	--	--	--
TP53	Tumor protein p53 [Source:HGNC Symbol;Acc:HGNC:11998	+	--	--	--	--	--
APP	Amyloid beta precursor protein [Source: HGNC Symbol; Acc: HGNC:620	--	+	--	--	--	--
DAB1	DAB1, reelin adaptor protein [Source: HGNC Symbol; Acc: HGNC:2661	--	+	--	--	--	--
PINK1	PTEN-induced putative kinase 1 [Source: HGNC Symbol; Acc: HGNC:14581	--	+	--	--	--	--
RELN	Reelin	literature review + miRNA-gene regulatory network					

**Table 4 cancers-15-03158-t004:** Identifying five Key miRNAs by considering two parameters (degree and betweenness centralities).

Label	Degree	Betweenness
hsa-mir-221-3p	4	5682.13
hsa-mir-30a-5p	4	2373.43
hsa-mir-15a-5p	3	3710.08
hsa-mir-130a-3p	3	3589.18
hsa-let-7b-5p	2	2523.74

**Table 5 cancers-15-03158-t005:** The expression status of eleven genes identified in GBM disease. Six genes represent a significantly increased expression in the GBM state, whereas four genes were downregulated. All eleven specific genes were altered during the primary stage of the tumor. “+”: presence; “--”: absence; ns: nonsignificant.

Gene Name	Non-Tumor	GBM	Pairwise *t*-Test (GBM-Non-Tumor) p.adj (*p*-Value with Bonferroni Correction)	Primary	Secondary	Recurrent
UBC	--	+	1.8 × 10^−3^	+	--	--
HDAC 1	--	+	7.8 × 10^−18^	+	--	--
CTNNB1	--	+	6.0 × 10^−3^	+	--	--
TRIM28	--	+	1.1 × 10^−3^	+	--	--
CSNK2A1	--	+	6.9 × 10^−1^ (ns)	+	--	--
RBBP4	--	+	3.2 × 10^−5^	+	--	--
TP53	--	+	1.6 × 10^−13^	+	--	--
APP	+	--	1.2 × 10^−3^	+	--	--
DAB1	+	--	4.0 × 10^−4^	+	--	--
PINK1	+	--	2.9 × 10^−10^	+	--	--
RELN	+	--	5.7 × 10^−8^	+	--	--

**Table 6 cancers-15-03158-t006:** The application of multiple criteria in the metabolic pathway analysis.

Result	Visualization Methods	Enrichment Method	Topology Analysis	Reference Metabolome	Pathway Library
**1**	Scatter plot	Hypergeometric test	Relative-betweenness centrality R-b C	All compounds in the selected pathway library	Homo sapiens (KEGG)
**2**	Scatter plot	Hypergeometric test	Out-degree Centrality O-d C	All combinations in the selected pathway library	Homo sapiens (KEGG)
**3**	Scatter plot	Hypergeometric test	Relative-betweenness centrality	All compounds in the selected pathway library	Homo sapiens (SMPDB)
**4**	Scatter plot	Hypergeometric test	Out-degree Centrality	All combinations in the selected pathway library	Homo sapiens (SMPDB)

**Table 7 cancers-15-03158-t007:** The final results from the metabolic pathway analysis. Further discussion of two items from the KEGG database (nitrogen metabolism, alanine, aspartate, and glutamate metabolism) and three from the SMPDB database (alanine metabolism, aspartate metabolism, and malate-aspartate shuttle) due to vitality. “+”: accept; “--”: reject.

KEGG Database						SMPDB Database					
Result 1	R-b C Impact	FDR	Result 2	O-d C Impact	FDR	Result 3	R-b C Impact	FDR	Result 4	O-d C Impact	FDR
	Final Decision (FD)			Final Decision (FD)			Final Decision (FD)			Final Decision (FD)	
Nitrogen metabolism	1	0.043213	Arginine biosynthesis	0.8125	1.90 × 10^−7^	Alanine metabolism	1	0.010641	Malate-aspartate shuttle	0.63333	0.013128
	FD: +			FD: --			FD: +			FD: +	
Phenylalanine, tyrosine, and tryptophan biosynthesis	1	0.12885	Alanine, aspartate, and glutamate metabolism	0.75	1.73 × 10^−7^	Trehalose degradation	0.84211	0.18355	Phosphatidylcholine biosynthesis	0.56707	0.00011577
	FD: --			FD: +			FD: --			FD: --	
Synthesis and degradation of ketone bodies	0.86667	0.18716	Valine, leucine, and isoleucine biosynthesis	0.75	8.82 × 10^−5^	Aspartate metabolism	0.8	0.0044894	Transfer of acetyl groups into mitochondria	0.54167	0.010641
	FD: --			FD: --			FD: +			FD: --	
Alanine, aspartate, and glutamate metabolism	0.81732	1.73 × 10^−7^	Nitrogen metabolism	0.75	0.043213	Glycerol phosphate shuttle	0.7619	0.3023	Ammonia recycling	0.49306	0.00011577
	FD: +			FD: +			FD: --			FD: --	
One-carbon pool by folate	0.80793	0.46957	Phenylalanine, tyrosine, and tryptophan biosynthesis	0.75	0.12885	Malate-Aspartate Shuttle	0.71429	0.013128	Cardiolipin biosynthesis	0.49057	0.013128
	FD: --			FD: --			FD: +			FD: --	

**Table 8 cancers-15-03158-t008:** The application of multiple criteria in the joint pathway analysis.

Result	Enrichment Method	Topology Measure	Integration Method
1	Hypergeometric test	Degree centrality	Combined score
2		Betweenness centrality	
3		Closeness centrality	

**Table 9 cancers-15-03158-t009:** The final results from joint pathway analysis. The two items observed in all three centralities (citrate cycle and arginine biosynthesis) were selected and further discussed. “+”: presence; “--”: absence.

Title	Degree	Betweenness	Closeness
Alanine, aspartate and glutamate metabolism	+	+	--
Citrate cycle (TCA cycle)	+	+	+
Arginine biosynthesis	+	+	+
Synthesis and degradation of ketone bodies	+	--	+
Pyruvate metabolism	+	--	+
Purine metabolism	+	+	--
Glutathione metabolism	+	+	--
Pyrimidine metabolism	+	+	--
Glycolysis or gluconeogenesis	--	+	+

**Table 10 cancers-15-03158-t010:** The summary of all results obtained from the pathway enrichment analysis at different pathway levels and joint pathway analyses.

Enrichment Analysis			Pathway Analysis	Joint Pathway Analysis
Eleven Genes	Five miRNAs	182 Metabolites		
Mitophagy	Fatty acid biosynthesis	Aminoacyl-tRNA biosynthesis	Nitrogen metabolism	Citrate cycle (TCA cycle)
Wnt signaling pathway	Galactose metabolism	Arginine biosynthesis	Alanine, aspartate and glutamate metabolism	Arginine biosynthesis
	Mucin-type O-glycan biosynthesis	Alanine, aspartate and glutamate metabolism	Malate-aspartate shuttle	
	Autophagy	Glutamate metabolism		
		Urea cycle		
		Arginine and proline metabolism		

**Table 11 cancers-15-03158-t011:** The metabolites and SNPs from the mGWAS-Explorer database.

Metabolite	SNP
HDL	rs111929233, rs7298751
N6-acetyllysine	rs12602273, rs12603869, rs12945970, rs12947788, rs12949655, rs12951053, rs1642782, rs17881556, rs1794284, rs2078486, rs5819163
Cholesterol	rs35608584, rs111929233
Formate	rs17520463
N, N-Dimethylglycine/Xylose	rs41450451
X2.piperidinone	rs75787097, rs75524270, rs79232054, rs145435197, rs74901488, rs117235978

## Data Availability

All data are uploaded as Appendix A.

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
