# Peer review of "Integrating Multi-Omics Analysis for Enhanced Diagnosis and Treatment of Glioblastoma: A Comprehensive Data-Driven Approach"

_cancers, 2023, doi:10.3390/cancers15123158_

Round 1
Reviewer 1 Report
This is a comprehensive bioinformatic analysis integrating molecular results from databases and literature search to identify novel glioblastoma target molecules.
The authors highlighted novel genes and miRNAs that seem to be important for glioblastoma development. Specifically, they are identified metabolites and metabolic profiles to be crucial. They are right to underline that those can become very important for monitoring disease progression and response to therapy and can give rise for biomarker development.
The authors explain their step-wise analysis very well, although I would suggest to concise a bit. E.g. discussion is very detailed but stretched.
I have no major concerns and think that the content is really interesting for glioblastoma research and future therapy development.
Author Response
This is a comprehensive bioinformatic analysis integrating molecular results from databases and literature search to identify novel glioblastoma target molecules.
The authors highlighted novel genes and miRNAs that seem to be important for glioblastoma development. Specifically, they are identified metabolites and metabolic profiles to be crucial. They are right to underline that those can become very important for monitoring disease progression and response to therapy and can give rise for biomarker development.
Q1: The authors explain their step-wise analysis very well, although I would suggest to concise a bit. E.g., discussion is very detailed but stretched. I have no major concerns and think that the content is really interesting for glioblastoma research and future therapy development.
A1: Thanks for taking the time to provide your valuable comments. We have done our best to make a concise discussion of the results. But the number of pathways and their cross talk are so extensive and caused the discussion to be little bit long. If we plan to make it more shorten the quality of paper would significantly dropped.
Reviewer 2 Report
The study has performed a multi-omics integrative analysis to identify key omics features, including gene expression, miRBAs, metabolites as well as signaling pathways that are associated with the progression of GMB.
The datasets that the authors used to perform the integrative analysis have not been clearly outlined in the paper. It’s better to summarize all the data sources used for the proposed analysis in a table. Why not use the TCGA data that has been widely adopted for multiomics analysis for GBM? The copy number variations and DNA metalations are also important omics features, but they have not been analyzed in the study.
One main concern is that survival outcomes have been ignored in the integrative analysis, so it is not convincing to many readers that the identified omics features are indeed critical for understanding the prognosis of GBM. More importantly, a large number of statistical methods for integrative analysis are ignored. Please refer to the review paper Wu et al. (PMCID: PMC6473252) for more details. A discussion of statistical integrative methods, especially those based on variable selection methods, should be included.
The study ignored the stability analysis of the identified omics features that are essential to claim the “importance” of identified omics features, given the heterogeneity of cancers.
Author Response
The study has performed a multi-omics integrative analysis to identify key omics features, including gene expression, miRBAs, metabolites as well as signaling pathways that are associated with the progression of GMB.
Q1: The datasets that the authors used to perform the integrative analysis have not been clearly outlined in the paper. It’s better to summarize all the data sources used for the proposed analysis in a table.
A1: Please accept our sincere thanks for the constructive suggestion. In this type of study, articles related to the databases used are usually cited, which is what we did. In order to provide more clarity on all sources, we summarized all sources in a table and added them to the materials and method (Page 5). Moreover, to summarize the sources of data, we also presented the bioinformatics steps used in figure 1 as a graph. (Page 6).
Q2: Why not use the TCGA data that has been widely adopted for multiomics analysis for GBM? The copy number variations and DNA metalations are also important omics features, but they have not been analyzed in the study.
A2: Thank you very much for your thoughtful inquiry. TCGA is undoubtedly one of the most important sites for cancer research; however, our purpose here was different. TCGA data are classified into seven categories based on a reference published in Nature in 2022 (Big data in basic and translational cancer research. Nat Rev Cancer. 2022 Nov;22(11):625-639. doi: 10.1038/s41568-022-00502-0. Epub 2022 Sep 5. PMID: 36064595; PMCID: PMC9443637.)
- Gene expression, 2. DNA mutations, 3. DNA methylation, 4. chromatin accessibility, 5. copy number alteration (CNA), 6. protein expression, 7. histopathology images.
In this study, we investigate the status of proteins involved in glioblastoma disease based on centrality parameters and within a comprehensive and protein-protein interaction network. It is not sufficient to focus on increasing or decreasing gene expression in the study of cancer for two reasons. One is that there may be several genes that are very important. However, their regulatory mechanisms are not at the level of transcription but at the level of translation. They occur after translation rather than transcription. Second, a gene may not cause changes in gene or protein expression. However, it may play a crucial role in the pathogenesis of glioblastoma and drug resistance development. The genes involved in glioblastoma are identified when your research approach is primarily to identify all genes involved in the disease (regardless of their mechanism of action) and draw connections between them, and then evaluate their centrality. Following the identification of the most important genes in the pathogenicity network, the next step was to investigate the mechanisms through which these genes exert their influence. In discussing DNA mutations, DNA methylation, chromatin accessibility, and copy number alteration (CNA), it should be remembered that these mechanisms are defined in terms of their effects on the expression and function of a gene in cancer. Since this gene was present in one of the several important databases used in this study, if it played a role in cancer, it was included in the analysis. Further, the databases used for collecting data in this study are important databases for identifying the targets involved in the disease for the purpose of computational drug design and the application of artificial intelligence and machine learning to drug discovery (The reference is to the following article, which was published in 2022: Application of Artificial Intelligence and Machine Learning in Drug Discovery. Methods Mol Biol. 2022; 2390:113-124. doi: 10.1007/978-1-0716-1787-8_4. PMID: 34731466.). (Page 11, lines 252-256)
However, this study used the TCGA tool to determine the correlation between the identified genes (GlioVis). By analyzing the TCGA database, we can evaluate the expression patterns of important genes identified as a result of the network to each other and determine their impact on disease development mechanisms. For the purpose of analyzing survival, OSgbm, which utilizes TCGA, GEO, and CGGA, was employed.
Q3: One main concern is that survival outcomes have been ignored in the integrative analysis, so it is not convincing to many readers that the identified omics features are indeed critical for understanding the prognosis of GBM.
A3: Our sincere thanks go out to you for your informative question. Based on the results of network analysis, we performed survival analysis on proteins that had previously been proven to be important. (Page 15, lines 322-325 and page 17, lines 345-348)
Q4: More importantly, a large number of statistical methods for integrative analysis are ignored. Please refer to the review paper Wu et al. (PMCID: PMC6473252) for more details. A discussion of statistical integrative methods, especially those based on variable selection methods, should be included.
A4: Regarding your enlightening query, I sincerely appreciate it. This approach is appropriate when you wish to integrate different types of data together and then analyze them. As part of the integration process, a sequential approach has been taken so that the protein network of different strategies is analyzed first. To identify important miRNAs, a regulatory network was constructed from the obtained proteins. Following that, metabolic networks were analyzed separately and in conjunction with the identified important proteins. Additionally, the type of enrichment method used is specified in the web server method. Since the approach used in this study is not machine learning, the discussion of statistical integrative methods, variable selection methods (supervised, unsupervised, and semi-supervised), and Bayesian variable selection is not meaningful. (Page 4, lines 132-141)
The strategy that you are referring to was used in our previous paper which was different from the current project (Pneumococcal Disease and the Effectiveness of the PPV23 Vaccine in Adults: A Two-Stage Bayesian Meta-Analysis of Observational and RCT Reports. Sci Rep. 2018 Jul 23;8(1):11051. doi: 10.1038/s41598-018-29280-2. PMID: 30038423; PMCID: PMC6056566).
Q5: The study ignored the stability analysis of the identified omics features that are essential to claim the “importance” of identified omics features, given the heterogeneity of cancers.
A5: Thank you very much for your insightful question. Stability and validity are discussed in order to assess the reliability of the selected set of features and variables. A measure of stability is the consistency of the obtained features across different data sets and platforms. In order to select features, one must take into consideration whether the model is supervised, unsupervised, or semi-supervised. Since supervised, unsupervised, and semi-supervised were not used in this study, the selection of features didn't occur, so checking stability is not meaningful. (Page 4, lines 142-148)
Round 2
Reviewer 2 Report
I thank the authors for addressing all the comments.
Author Response
The authors appreciate the respected reviewer positive feedbacks to improve our paper.